# Does Resveratrol Improve Metabolic Dysfunction-Associated Steatotic Liver Disease (MASLD)?

**DOI:** 10.3390/ijms25073746

**Published:** 2024-03-27

**Authors:** Kamila Kasprzak-Drozd, Przemysław Niziński, Paulina Kasprzak, Adrianna Kondracka, Tomasz Oniszczuk, Agata Rusinek, Anna Oniszczuk

**Affiliations:** 1Department of Inorganic Chemistry, Medical University of Lublin, Chodźki 4a, 20-093 Lublin, Poland; kamilakasprzakdrozd@umlub.pl (K.K.-D.); agatarusinek997@gmail.com (A.R.); 2Department of Pharmacology, Medical University of Lublin, Radziwiłłowska 11, 20-080 Lublin, Poland; przemyslaw.nizinski@umlub.pl; 3Department of Conservative Dentistry with Endodontics, Medical University of Lublin, Chodźki 6, 20-093 Lublin, Poland; paulina.kasprzak87@gmail.com; 4Department of Obstetrics and Pathology of Pregnancy, Medical University of Lublin, Staszica 16, 20-081 Lublin, Poland; adriannnakondracka@umlub.pl; 5Department of Thermal Technology and Food Process Engineering, University of Life Sciences in Lublin, Głęboka 31, 20-612 Lublin, Poland; tomasz.oniszczuk@up.lublin.pl

**Keywords:** resveratrol, MASLD, polyphenols, nutraceuticals, metabolic syndrome, functional food, lipids, oxidative stress, antioxidants

## Abstract

Metabolic dysfunction-associated steatotic liver disease (MASLD) is influenced by a variety of factors, including environmental and genetic factors. The most significant outcome is the alteration of free fatty acid and triglyceride metabolism. Lipotoxicity, impaired autophagy, chronic inflammation, and oxidative stress, as well as coexisting insulin resistance, obesity, and changes in the composition of gut microbiota, are also considered crucial factors in the pathogenesis of MASLD. Resveratrol is a polyphenolic compound that belongs to the stilbene subgroup. This review summarises the available information on the therapeutic effects of resveratrol against MASLD. Resveratrol has demonstrated promising antisteatotic, antioxidant, and anti-inflammatory activities in liver cells in in vitro and animal studies. Resveratrol has been associated with inhibiting the NF-κB pathway, activating the SIRT-1 and AMPK pathways, normalizing the intestinal microbiome, and alleviating intestinal inflammation. However, clinical studies have yielded inconclusive results regarding the efficacy of resveratrol in alleviating hepatic steatosis or reducing any of the parameters found in MASLD in human patients. The lack of homogeneity between studies, low bioavailability of resveratrol, and population variability when compared to animal models could be the reasons for this.

## 1. Introduction

In recent years, metabolic dysfunction-associated steatotic liver disease (MASLD) has received considerable attention in research studies and clinical practice [1]. The disease is recognized on imaging or histology when more than 5% of the weight of the liver is lipid material and the exclusion of other causes of secondary steatosis, such as heavy alcohol consumption, hereditary disorders and steatogenic drugs [2]. Unified global approaches to nomenclature and disease definition are crucial for increasing disease awareness, identifying those at risk, and facilitating diagnosis and access to care. It has been established for many years that being overweight or obese is linked to hepatic steatosis, hepatocyte injury, liver inflammation, and fibrosis. This was formally recognized as ‘non-alcoholic steatohepatitis’ in 1980 by Jurgen Ludwig [3]. Subsequently, the term ‘non-alcoholic fatty liver disease’ (NAFLD) was used to describe the histological spectrum from steatosis to steatohepatitis. However, the term NAFLD has limitations as it relies on exclusionary confounder terms and uses potentially stigmatizing language. Therefore, it has been replaced with the term ‘metabolic dysfunction-associated steatotic liver disease’ (MASLD) [3,4]. Younossi et al.’s [5] meta-analysis found the worldwide prevalence of MASLD to be 25.24%, with the Middle East and South America having the highest prevalence and Africa having the lowest. Other published meta-analyses report global MASLD prevalence figures of 29.8% [6] and 32.4% [7]. Data from Europe indicated that the incidence of MASLD in adults is 26.9%, with a greater incidence in men compared to women (32.8% vs. 19.6%) and in patients with metabolic syndrome and its components compared to those without (75.3% vs. 17.9%) [8]. MASLD is forecast to rise considerably across several global regions by 2030 if present trends persist [9].

MASLD is the first step in the development of irreversible changes in the liver parenchyma leading to cirrhosis (about 1/3 of cases of MASLD develop steatohepatitis, and 15% of these may progress to cirrhosis), while on the other hand, MASLD per se may be a risk factor for the development of MASLD the first step in the development of irreversible changes in the liver parenchyma leading to cirrhosis (about one-third of cases of MASLD develop steatohepatitis and 15% of these may progress to cirrhosis), while on the other hand MASLD per se may be a risk factor for the development of premature atherosclerosis, type 2 diabetes (T2DM) and hepatocellular carcinoma [10,11,12]. In 2010, it was estimated that 4% of all deaths worldwide were due to major liver diseases such as cirrhosis and hepatocellular carcinoma [13]. As of 2020, there are no FDA-approved pharmacological therapies for MASLD, representing a significant unmet therapeutic need [14]. MASLD is now considered a multisystem disease, which is associated with an increased risk of type 2 diabetes mellitus, cardiovascular disease, and chronic kidney disease. Furthermore, it is analyzed as the hepatic manifestation of the metabolic syndrome. As mentioned, there are no effective pharmacological therapies for MASLD. Therefore, the prevention and management of the illness involves lifestyle changes, such as weight reduction through a healthy diet and physical activity [1]. Problems with drug therapy of MASLD increased interest in active food constituents as new approaches to its treatment. Natural products play an important and irreplaceable role in drug development and design [15]. Currently, more and more natural products have shown great therapeutic potential in various diseases, including MASLD [16]. Resveratrol is an important natural constituent with therapeutic potential. It has shown pleiotropic potential due to its anti-inflammatory and antioxidant properties, as well as protective actions against age-related disorders. Furthermore, resveratrol has been shown to improve cellular and vascular function as well as metabolic health in obese patients by ameliorating glucose control and insulin sensitivity. The beneficial effects of this polyphenol on obesity-related complications are largely due to its ability to mimic the effects of calorie restriction [17,18,19,20,21,22]. This review aims to analyze the effects of resveratrol on hepatic steatosis, including its efficacy in in vitro studies, animal studies, and clinical trials.

## 2. Pathogenesis of MASLD

The mechanisms of MASLD development and progression are complex and not yet fully understood. Among the non-genetic risk factors for MASLD, certain conditions appear to be the most significant. They include but are not limited to increased body mass index (i.e., overweight and obesity), alterations in serum lipid profile (hypercholesterolemia, hypertriglyceridemia, elevated free fatty acids levels), as well as insulin resistance or T2DM [18,23]. There is also growing evidence of the correlation between sleep disorders and MASLD. Recent studies show that obstructive sleep apnoea syndrome (OSAS) and insomnia could be associated with MASLD development [19]. Hypothyroidism and thyroid hormone alterations are also considered emerging risk factors of MASLD. Many of the above-mentioned potential reasons for MASLD occurrence are often correlated with poor health habits, in particular, a sedentary lifestyle and general physical inactivity or high consumption of processed food and sweetened drinks, particularly those containing high amounts of fructose [18]. As it was mentioned earlier, many natural products are currently in the scope of potential therapeutical effects for liver steatosis. In Figure 1, a number of potentially MASLD-improving nutraceuticals connected with related causes of the disease are shown.

In order to address the issues of MASLD pathogenesis, some theories have been formulated, taking into consideration the complexity of liver diseases with regard to multiple parallel processes in the human organism, as well as the genetic and environmental factors that might be implicated in the development and progression of the disorder. A recently proposed and continuously evolving “multiple-hit” hypothesis describes potential mechanisms that underlay MASLD pathogenesis and progression [24]. Below, these pathways are delineated, and their contribution to MASLD development is discussed.

### 2.1. Metabolism of Lipids and Insulin Resistance

Alterations in pathways involved in lipid metabolism could lead to disruption of lipid homeostasis. Notably, three main causes of excessive lipid accumulation in the liver have been described [23]. High-fat absorption from dietary products, high levels of hepatic lipid synthesis, and exaggerated amounts of fat in adipose tissue have been proposed as crucial factors. All the aforementioned contribute to the increased uptake of free fatty acids (FFAs) by the hepatocytes. FFAs derive either from nutritional products after digestion or from de novo lipogenesis (DNL) in the liver, as well as via lipolysis from the adipose tissue [23,24]. Exaggerated serum FFA levels play a vital role in liver steatosis by initiating excessive triglycerides (TG) synthesis and promoting their accumulation in the liver tissue rather than enabling transport to the subcutaneous adipose tissue for storage.

Apart from an increased flux of lipids to the liver, in MASLD, an impairment of lipid efflux to peripheral tissues is also confirmed. Alterations in lipid metabolic pathways could result in an imbalance of high-, low- and very low-density (HDL, LDL, and VLDL, respectively) lipoproteins transport and fatty acids oxidation, which are the most important routes of eliminating excessive hepatic TG and cholesterol [25]. Elevated levels of FFAs can also be a result of coexisting insulin resistance, which is considered one of the key factors of MASLD development. The DNL rate can be increased by activation of a number of transcription factors, where isoform 1c of sterol regulatory element-binding protein-1 (SREBP-1c) and carbohydrate response element binding protein (ChREBP) play crucial roles. Since SREBP-1c is regulated by insulin receptor substrate 2 (IRS-2), in states of insulin resistance, IRS-2 is down-regulated. This brings about overexpression of SREBP-1c and up-regulation of DNL [24]. Unlike SREBP-1c, ChREBP activation is mediated by glucose, and it also takes part in providing substrates to FFAs and TG synthesis [26]. Additionally, insulin has a potent influence on inhibiting adipose tissue lipolysis. If a cell’s response to insulin is impaired, an influx of FFAs to the liver increases.

### 2.2. Lipotoxicity, Oxidative Stress and Inflammatory Processes

Hepatic lipotoxicity is one of the most important events in the development and progression of MASLD. This phenomenon results from excessive FFA influx to liver tissue. Therefore, when hepatocytes are unable to efficiently convert FFAs into TG, it results in the continuous production of reactive oxygen species (ROS). Notably, saturated FFAs seem to be more hepatotoxic than unsaturated molecules. In vitro study results show that saturated palmitic or stearic acids are more detrimental to liver cells than monounsaturated species: oleate and palmitoleate [27]. Indeed, ROS might induce oxidative stress and the malfunction of endoplasmic reticulum (ER) or mitochondria [28]. Oxidative stress is perceived as one of the key features in the initiation and development of liver steatosis. In MASLD, mitochondria are the most important source of ROS due to increased fatty acid oxidation because of FFA overload [29].

Oxidative stress can also activate an inflammatory signaling cascade and subsequently induce overexpression of many pro-inflammatory bioactive compounds, including interleukin-6, interleukin-1β, tumor necrosis factor-α and transforming growth factor-β. In addition, pathways of cytochrome P450, lipoxygenase (LOX) and cyclooxygenase (COX), which are known as pro-oxidant and pro-inflammatory systems, are also implicated in oxidative stress processes. To sum up, the activity of these proteins results in a vicious cycle of oxidative stress and chronic inflammation, which seems to be the reason for the histopathological and biochemical changes in the pathogenesis of MASLD [30,31,32,33]. Moreover, in individuals with liver steatosis, the activities of a number of antioxidant endogenous enzymes (e.g., catalase, superoxide dismutase) and exogenous ROS scavengers (e.g., vitamin C and vitamin E) are reduced [28].

### 2.3. Gut Microbiota

The role of gut microbiota in the development and progression of MASLD has gained significant attention in recent years. More than 25 years ago, the hypothesis of the ‘gut–liver axis’ was proposed, and over time, there is increasing evidence of the role of gut microbiota in liver function. Specific bacteria and other microorganisms seem to be involved in many pathways influencing the development and progression of MASLD [23]. Accordingly, impaired function of gut microbiota and distinct changes in abundance or deficiency of certain bacteria species could lead to alterations in the secretion of microbial metabolites [34]. For instance, one of the most well-known disrupting factors of microbiota composition in MASLD is excessive intake of fructose. It has been reported that a high fructose (HF) diet in mice results in a decrease of diversity in the composition of gut microbiota, particularly in increasing Firmicutes/Bacterioidetes ratio. Elevated levels of Firmicutes have been linked with a higher risk of diabetes or obesity, and it may lead to the development of MASLD. Further observations suggest that gut microbiota alterations can lead to enhanced expression of some pro-inflammatory molecules (TNF-α, IL-6, IL-1β) and increased T-helper type-1 cells [35,36]. At the microbiota genera level, increased abundance of, i.e., *Escherichia*, *Dorea*, *Peptoniphylus* and decreased *Anaerosporobacter*, *Coprococcus*, *Eubacterium*, *Faecalibacterium*, *Prevotella* has been reported in MASLD individuals when compared to healthy subjects [37]. In turn, studies in children suggest that elevated levels of Bacterioidetes, Proteobacteria, Enterobacteriaceae, Gammaproteobacteria, and lowered Firmicutes phyla can be significantly associated with liver steatosis [38]. A recent study by Wei et al. also suggested that elevated levels of *Klebsiella*, *Clostridium*, *Enterococcus*, and *Bacteroides* genera might cause microbiome imbalance and injuries of gut tissue [39]. A comprehensive review of microbiome alterations in liver steatosis was neoterically published elsewhere [37]. In both physiological and pathological conditions, microbe-derived molecules could affect many receptors and metabolic pathways in the host organism, including lipids metabolism [34].

It has been confirmed that the gut microbiota plays a crucial role in the metabolic reactions of bile acids, primarily by converting primary acids to biologically active secondary derivatives. Alterations in gut microbiota composition can lead to dysfunction of TGR5 and FXR signaling pathways [40,41]. Therefore, the gut microbiome may be of great importance in bile acid metabolism and the indirect regulation of carbohydrate and lipid metabolisms. They play a vital role in maintaining gut microbiota’s proper composition and function. Short-chain fatty acids (SCAs) are essential nutrients in metabolic and immunological processes and in the expansion of adipose tissue in the colon. Research on butyrate, which belongs to SCAs, has shown that a decrease in the rate of its synthesis by gut microbiota may lead to an increase in gut permeability, thereby increasing the risk of lipopolysaccharide (LPS) and bacterial translocation to the systemic circulation [42]. Butyrate also contributes to GLP-1 signaling by upregulating GLP-1 receptor expression, thereby helping to improve GLP-1-mediated beneficial effects in MASLD [43]. Ethanol is a small molecule that, when present in elevated levels due to certain gut microbiota alterations, could potentially impact the progression of MASLD. Excessive concentrations of ethanol and its derivatives in the intestines may lead to damage to epithelial cells, which in turn could increase intestinal permeability to LPS and other toxic molecules [44,45,46].

### 2.4. Autophagy and Thyroid Hormones

Autophagy is a process aiming towards the maintenance of cellular homeostasis by degrading damaged or dysfunctional cellular components, including organelles and proteins. Another purpose of autophagy is the recycling of small molecules (carbohydrates or amino acids) that can be readily reused in processes of more complex compound formation. Autophagy is reduced in patients with MASLD. This situation leads to impairment of TG formation and efflux from the liver [47,48,49]. In vitro and in vivo studies show that inhibition of autophagy in the liver results in decreased lipid oxidation and increased steatosis; hence, it is highly probable that this is the main mechanism for converting stored TG into FFAs in the liver [50]. Thus, the failure of liver cells to utilize and export excess amounts of FFA is associated with an increased rate of hepatocyte apoptosis (deemed ‘lipoapoptosis’) [51,52].

Studies have demonstrated that thyroid hormones (TH) (notably, triiodothyronine (T3) and thyroxine (T4)) contribute to lipid metabolism. A number of metabolic pathways in the liver, including lipogenesis, lipid oxidation and gluconeogenesis, are partially regulated by TH. In vitro studies revealed that TH significantly increases hepatic autophagy processes; thus, TH-mediated autophagy seems to be one of the most important pathways in the mobilization of hepatic TG and in releasing FFAs for mitochondrial β-oxidation [53,54,55].

### 2.5. Genetic and Epigenetic Factors

Genetic factors can play a significant role in an individual’s susceptibility to MASLD. Indeed, several genetic variants contributing to the development of MASLD were described by genome-wide association studies (GWAS) [56]. To date, one of the most interesting targets of such study is patatin-like phospholipase domain-containing 3 (*PNPLA3*), also called adiponutrin [57]. A specific variant of this gene, known as rs738409 or I148M, is strongly associated with MASLD [58]. There are also investigations of the potential role of *GCKR*, *TM6SF2*, *HSD17B13*, *MB0AT7*, *PPP1R3B*, *IRGM* and *LPIN1* genes and their variants in the development and progression of MASLD. Accordingly, an extensive review of genetic predictors of MASLD has been recently published by Jonas and Schürmann [56]. It seems justified to seek a link between genetic, epigenetic, and environmental factors in the pathogenesis of MASLD. Crucially, understanding genetic predisposition to liver steatosis could be helpful in the prediction and diagnosis of MASLD.

## 3. Resveratrol

Since MASLD is strongly associated with the development of obesity and type-2 diabetes, a healthy diet and physical activity are the cornerstones of treatment. However, the main limitation of this kind of therapy is infrequent adherence, mainly due to emotional factors, stress and difficulties in modifying dietary patterns. Thus, researchers are constantly looking for alternatives that could improve the outcomes of the approaches currently used for MASLD treatment. Much attention has been paid to polyphenols, a heterogeneous group of plant-derived compounds that are naturally present in the diet. Among them, resveratrol has been one of the most studied ones. This polyphenol has been proposed as an energy restriction mimetic compound (without limiting the caloric intake). Therefore, resveratrol has been extensively studied as a potential anti-obesity agent [17,58].

### 3.1. Chemical Overview and Sources of Resveratrol

Nowadays, certain polyphenolic compounds (secondary plant metabolites) are of increasing research interest. These are a wide and heterogenous group of plant-derived molecules [59]. Two important subgroups can be distinguished, i.e., flavonoids and phenolic acids. The stilbens are another set of plant metabolites of investigative interest [60,61]. Stilbenes contain two phenyl compounds connected by a two-carbon methylene bridge. Most stilbenes in plants act as antifungal phytoalexins. These are compounds that are usually only synthesized in cases of response to injury or infection [62]. The most researched due to its exciting pharmacological potential is resveratrol (3,4′,5-trihydroxystilbene) with a C6-C2-C6 carbon skeleton [63] and molecular mass of 228.24 g/mol [64]. The chemical structure of this compound is shown in Figure 2. The origin of this name is related to the Latin language. The prefix “res” means ‘which comes from’, “veratr” from the plant ‘*Veratrum*’, and the suffix “ol”, indicates that it contains hydroxyl ‘alcohol’ groups in its structure [63].

Resveratrol is found in more than 70 plants, most of them edible [65,66,67]. The most important and most obvious source of resveratrol is the grapevine [68]. Dark grapes contain more of this compound than light grapes. The Pinot noir variety holds the highest content of resveratrol [69,70]. Resveratrol content in grape skins reaches 50–100 mg/g in fresh weight [62]. This compound is also present in such plants as pomegranates, ground nuts [68], peanuts, pistachios, blueberries, bilberries [71], and Pinus sylvestris [72].

Several processed plant products also contain a significant amount of resveratrol. An example is red wine, where this compound reaches concentrations in the range of 0.1–14.3 mg/L. This fact has been suggested as a possible solution to the “French paradox”. This is the observation of an unanticipated low rate of heart disease among the inhabitants of southern France, who consume a lot of red wine despite the fact that their diet is abundant in unhealthy saturated fats [73].

Resveratrol occurs in two isomeric forms: cis- and trans-. Trans-resveratrol is a more common form that is characterized by higher biological activity and greater stability [64]. This compound undergoes a cycle of enterohepatic metabolism. Notably, as a result of the immediate uptake of this compound by enterocytes, it is metabolized, among others, to glucuronides (3-O-glucuronide and 4′-O-glucuronide) and sulfate conjugates (3-O-sulfate). These compounds can be deconjugated and reabsorbed or excreted in the feces, whereas this occurs after they are secreted back into the intestine. Comparatively low levels of resveratrol in the blood can be explained by the enterohepatic cycle and its rapid metabolism in the liver [66].

### 3.2. Mechanism of Resveratrol Action

Resveratrol exhibits a diverse range of physiological and biochemical activities, among others, antioxidant, anti-inflammatory, antiplatelet, anticoagulant, anti-dementia, and anti-aging effects. These mechanisms are pivotal in their advantageous impact. Experimental studies have found that resveratrol inhibits the oxidation of cell membrane lipids, hence protecting low-density lipoproteins (LDL) from oxidation, as well as increasing the concentration of high-density lipoproteins (HDL). It also manifests strong antioxidant properties through its influence on the formation of reactive oxygen species (ROS) [74]. Under physiological conditions, ROS are maintained at a certain level, and excess free radicals are typically removed by antioxidant enzymes such as superoxide dismutase (SOD), catalase (CAT) and glutathione peroxidase (GPH-Px). Nevertheless, the consequences of the homeostasis disorder between the antioxidants and the oxidative system may be an increase in the level of ROS. ROS or other free radicals may interfere with cell function by affecting transcription factors and the redox-sensitive signaling pathway. The nuclear factor associated with erythroid factor 2 (Nrf2) is a transcription factor that has a significant regulatory effect on oxidative states by inducing the expression of phase 2 antioxidant and detoxifying enzymes and related proteins. Studies suggested that resveratrol may influence Nrf2 as a cellular target, thereby controlling the process and signaling pathways linked with reducing oxidative stress [75] that which item involves the transcription of antioxidant genes, such as that involves the transcription of antioxidant genes such as SOD and CAT [76].

Resveratrol indirectly triggers autophagy via either the mTOR-dependent or TFEB-dependent pathway, whereby AMPK/SIRT1/Nrf2, ERK/p38, MAPK, and PTEN/Akt signaling pathways are involved. This promotes the synthesis of antioxidant molecules and the expression of corresponding genes [18,77].

Resveratrol can activate sirtuin-1 (SIRT-1) proteins [74]. SIRT1 gene expression regulates carbohydrate and lipid metabolism, thereby reducing fat accumulation and lowering the risk of visceral obesity. The SIRT1 gene also plays a role in apoptosis, differentiation and cellular aging. Consequently, reduced SIRT1 gene expression predisposes to metabolic syndrome development [78]. One of the most dependable outcomes of treatment with resveratrol is the growth in mitochondrial mass. SIRT1 advances mitochondria creation by means of deacetylation and activating PGC-1α (peroxisome proliferator-activated receptor-gamma coactivator), a lead regulator in the production of mitochondria. PGC-1α also collaborates with nuclear respiratory factors (NRF-1 and NRF-2) that are responsible for initiating the transcription of genes that contribute to the generation of mitochondria. PGC-1α can also be activated by AMPK (AMP-activated protein kinase), a significant metabolic sensor. SIRT1 is a crucial factor in the capability of moderate resveratrol doses to stimulate AMPK activation and enhance mitochondrial function [79]. In terms of its effect on reducing hepatic fat accumulation (decreased lipogenesis, increased fatty acid oxidation), activation of SIRT1 and AMPK improves metabolic lipid homeostasis [18]. Resveratrol supplementation decreases inflammation while minimizing hepatic steatosis via activation of the AMPKα-SIRT1 pathway, suppressing the nuclear factor kappa B (NF-κB) inflammatory pathway [80].

In cases of chronic liver damage, fibrogenesis arises within the liver as a protective response [81]. Resveratrol is renowned for its ability to reverse adverse processes. Studies have revealed that it reduces ALT and AST, TNF-α, and IL-6 mRNA levels, as well as the recruitment of Kupffer cells (CD68+) in the damaged liver [82]. Selected molecular mechanisms of resveratrol action on MASLD improvement are shown in Figure 3.

### 3.3. Effect of Resveratrol on Liver Condition—In Vitro Studies

The treatment options for MASLD are limited, which has led to the exploration of bioactive natural compounds to reduce hepatic steatosis. Studies on resveratrol have shown that it can lower lipids by inhibiting adipogenesis, increase lipid mobilization to provide a hepatoprotective effect, and improve insulin sensitivity [83]. Unfortunately, most studies have been conducted in vitro or in animal models, and the clinical trials that have been conducted are inconclusive.

HepG2 are hepatoma cell lines and have a high proliferation rate. They have an epithelial-like morphology and perform many differentiated hepatic functions. HepG2 lines are commonly used as in vitro alternatives to primary human hepatocytes in drug metabolism and hepatotoxicity studies. However, they have a major limitation: they express some metabolic activities at a lower level than hepatocytes. In a study conducted by Izdebska et al. [84] on HepG2 lines, resveratrol at doses of 10 and 20 μM after 24 h of treatment reduced the degree of steatosis and mitochondrial oxidative stress (induced by oleic/palmitic acid) in HepG2 cells. The same investigators confirmed that resveratrol at a dose of 20 μM can mitigate glucose-induced steatosis and improve mitochondrial function in HepG2 cells [85]. Using the same cells, other researchers demonstrated that resveratrol (1–10 μM) can protect against the production of reactive oxygen species (ROS) produced by oleic acid intervention [86]. Resveratrol also induced mitochondrial biogenesis and increased the expression of mitochondrial respiratory subunits as well as manganese superoxide dismutase. Moreover, it improved the synthesis of TNF-α and reduced levels of UCP-2.

Hepatic lipid accumulation promotes the development of MASLD in individuals with type 2 diabetes mellitus. Sirtuin-1 plays a crucial role in the pathogenesis of MASLD by regulating glycolipid metabolism. In a study by Yang et al. [87], the mechanisms of resveratrol action in hepatic lipid accumulation induced by type 2 diabetes mellitus were investigated. The authors found that a dose of 20 μM of resveratrol significantly reduced triglyceride deposition and lipid accumulation in high glucose-cultured HepG2 cells. Further research has linked this benefit to the upregulation of SIRT1, which inhibits farnesol X receptor acetylation. It is important to note that the hepatoprotective effect of resveratrol was lost when the farnesol X receptor was silenced.

Trepiana et al. [88] studied the potential of resveratrol metabolites (trans-resveratrol-40-O-glucuronide, trans-resveratrol-3-O-sulfate, trans-resveratrol-3-O-glucuronide, and dihydro resveratrol) in reducing hepatocyte fat accumulation. They developed an in vitro model by treating mouse AML12 hepatocytes with palmitic acid. During the experiment, hepatocytes were incubated with 1, 10, or 25 μM resveratrol or its metabolites. The findings showed that resveratrol and its metabolites at 1 μM prevented lipid accumulation induced by the saturated fatty acid in hepatocytes. This action was mainly determined by the inhibition of de novo lipogenesis. This study demonstrates that the low bioavailability of resveratrol is not an obstacle in this case because resveratrol metabolites possess delipidating effects like the parent compound. Results of the resveratrol administration on hepatic steatosis in in vitro models and its mechanisms of action are presented in Table 1.

### 3.4. Effect of Resveratrol on Liver Condition—Animal Studies

Animal studies have shown that resveratrol can improve the histological structure of the liver, normalize lipid profile and glycemia, and reduce circulating leptin. Additionally, it exhibits antioxidant and anti-inflammatory properties.

Another study discovered that resveratrol can decrease liver steatosis induced by a high-fat diet in animals. The study involved in vitro tests and research on a rat model. The study found that resveratrol decreased body weight, as well as concentrations of alanine and aspartate aminotransferase, glucose, insulin, and lipids. Furthermore, in an in vitro test, this polyphenol reduced lipid accumulation induced by palmitate acid in HepG2 cells and, modulated fatty acid oxidation and returned antioxidant capacity and respiratory function of mitochondria [92].

Heebøll et al. [93] used a high-fat diet to generate a male rat MASLD model. They tested the preventive and therapeutic potential of resveratrol by adding resveratrol to the high-fat diet from the first day of the study or after 1 week of the diet. The findings suggest that a weak hepatic benefit of resveratrol treatment was found in the prevention of steatosis only. This compound reduced the development of histological steatosis and partly triglyceride accumulation in the MASLD model, although the effects were moderate. Supplementation with this polyphenol for 13 weeks effectively protected against lipid over-accumulation in the liver cells and improved the serum lipid profile in a rat MASLD model caused by a high-fat and high-sucrose diet [94]. Similar effects were observed in rats fed a high-fat diet (30 mg/kg per day for 60 days). The reduction in body weight, transaminase, and insulin levels, as well as an improvement in plasma lipid profile, were reported. In addition, resveratrol administration enhanced hepatic lipid metabolism by down-regulating the expression of adipogenic genes (e.g., acetyl-CoA carboxylase) and exerted anti-inflammatory activity by reducing the expression of the TNF-α and nuclear factor kappa-light-chain-enhancer of activated B cells (NF-κB) and interleukin 6 (IL-6) [95].

Poulsen et al. [96] administered resveratrol (100 mg daily for 8 weeks) to rats that have consumed a high-fat diet. The authors observed a significant reduction in the degree of hepatic steatosis and the normalization of triglyceride levels. These effects were correlated with an increase in mitochondrial biogenesis in the liver and uncoupling protein-2 expression. In this way, the excess energy (from lipid over-accumulation) was dissipated as heat. In another rat model, oral resveratrol at a dose of 10 mg daily diminished the degree of induced hepatic steatosis. Moreover, in this study, resveratrol decreased serum levels of glucose and TNF-α and liver levels of nitric oxide synthase and malondialdehyde (MDA). In addition, the levels of superoxide dismutase, catalase, and glutathione peroxidase increased compared with the steatosis group [97].

The aim of the next study [98] was to evaluate the resveratrol effects on the MAF1 gene liver pathway using a mouse model. Male mice were divided into four groups. Mice in the first and second groups were fed with a standard diet or standard diet enriched with resveratrol (4 weeks, 30 mg/kg/day/60 days). Mice in other groups ate a high-fat diet or a high-fat diet with resveratrol (4 weeks, 30 mg/kg/day/60 days). The results showed that this polyphenol can decrease the epididymal adipose tissue weight, enhancing glycemic and lipid parameters. Moreover, the collagen deposition was reduced, and the catalase enzyme activity improved. The mRNA expression of key genes improved, suggesting the lipid profile enhancement in resveratrol-treated animals. The data indicate that a 4-week administration of resveratrol caused significant liver fat reduction in obese mice, and this beneficial effect might be coordinated by the MAF1. In a study conducted by Hosseini et al. [99], the authors explored the effect of resveratrol on the epigenetic regulation of Nrf2-Keap1 in vivo models of MASLD. Thirty six-week-old male C57/BL6 mice were randomly divided into three dietary groups: a standard chow diet (10 kcal% fat), a high-fat diet (55.9 kcal% fat), and a high-fat diet supplemented with 0.4% resveratrol. Research has shown that resveratrol administered for 16 weeks can reduce high-fat diet-induced methylation of the Nrf2 promoter in mice liver. Researchers have observed a decrease in the expression of lipogenesis-related genes (FAS and SREBP-1c) and a reduction in triglyceride concentration. These findings confirmed that resveratrol can reduce MASLD through the epigenetic modification of the erythroid factor-2 signaling.

Resveratrol and caloric restriction can activate SIRT-1 and induce autophagy; moreover, autophagy is induced by the SIRT1-FoxO signaling pathway, and it can be a critical protective mechanism against MASLD development. Ding et al. [100] compared the impact of resveratrol and caloric restriction on the hepatic lipid metabolism of male rats. In addition, the authors studied the interplay between resveratrol consumption, SIRT1 and autophagy. The results showed that resveratrol (200 mg/kg/18 weeks) and caloric restriction (30%/18 weeks) partially prevented hepatic steatosis and hepatocyte ballooning, increased the expression of SIRT1 and autophagy markers while decreasing endoplasmic reticulum stress markers in the liver and alleviated lipid metabolism disorder.

Research by Tiao et al. [101] showed that a maternal high-fat diet brought about hepatic steatosis and apoptosis in rats via the renin-angiotensin system. Resveratrol (dose 50 mg/kg/day, up to 120 days after birth), in contrast, protected against MASLD, regulated lipid metabolism, and limited oxidative stress and apoptosis in the liver via SIRT-1 and renin-angiotensin secretion.

The efficacy of resveratrol was tested in a rat model of MASLD with hyperuricemia due to the feeding of animals with a yeast-rich, high-fat diet with potassium oxonate. This polyphenol at a dose of 100 mg/kg/day administered for 12 weeks improved liver structure and function, in addition to uric acid excretion. It also decreased hepatic steatosis, oxidative stress, and inflammation by activation of the SIRT-1 way [102]. Other results were obtained in research conducted by Kessoku et al. [103]. These showed no beneficial effects on hepatic steatosis in the oral administration of resveratrol for 4 weeks at a dose of 2–20 mg/kg/day in a model of MASLD mice induced by a high-fat diet. In these studies, the polyphenol, however, decreased inflammation and fibrosis in a lipopolysaccharide-induced MASLD model by inhibiting the signal transducer and activator of the transcription 3 (STAT3) pathway and suppressing CD14 protein expression in liver cells. Moreover, in the case of mice with high-fat diet-induced steatosis of liver cells, resveratrol brought down hepatic lipid accumulation, as well as levels of circulating pro-inflammatory mediators (TNF-α, IL-6, and IL-1β). Treatment with resveratrol (30 mg/kg/day) for 60 days reduced the hepatic nuclear factor κB (NF-κB) inflammatory pathway by activating AMP-activated protein kinase α (AMPKα) phosphorylation and SIRT-1 content [80].

The aim of a study conducted by other researchers [104] was to develop the impact of 6 weeks of resveratrol administration alone, aerobic exercise alone, and aerobic exercise together with resveratrol administration on a MASLD rat model. Aerobic exercise or resveratrol alone has different improvements on glucose, lipid metabolism disorders, and adipocytokine disorders in MASLD rats, but the greatest improvements were obtained with combined exercise and resveratrol administration.

Guo et al. [105] demonstrated that resveratrol can act against MASLD by modulating intestinal microbiota in mice. The authors aimed to investigate the beneficial effects of the main microbial metabolites from resveratrol on lipid metabolism. The mice were fed a high-fat diet and injected with resveratrol, 3-hydroxyphenyl propionic acid, and 4-hydroxyphenyl propionic acid in doses of 100 µM for 13 weeks. For intraperitoneal injection (i.p.), these compounds were dissolved in PEG400 and then diluted with saline to obtain 20% (*v*/*v*) PEG solutions. The results suggested that these three phenolic compounds effectively reduced liver weight and body weight, improved hepatic steatosis, and alleviated systemic inflammation in MASLD mice. In addition, the results showed that 3-hydroxyphenyl propionic acid and 4-hydroxyphenyl propionic acid altered the expression of cholesterol influx and efflux genes to a stronger extent than resveratrol.

Significant improvements in hepatic steatosis and lipid and glucose metabolism have also been observed in various animal model studies after treatment with resveratrol. The potential mechanisms of action are diverse. However, most involve SIRT-1, resulting in improved mitochondrial function. This action contributes to fatty acid oxidation, dissipation of surplus energy in the form of heat, and less ROS production. In addition, a reduction in the expression of adipogenic-related genes and pro-inflammatory pathways was observed. The results of the resveratrol administration on hepatic steatosis in animal models and its mechanisms of action are shown in Table 2.

### 3.5. Effect of Resveratrol on Liver Condition—Clinical Trials

Resveratrol’s effects on patients with MASLD have only been studied in a limited number of clinical trials. The experimental procedures used in these trials varied in terms of duration (8 weeks–6 months), dose of resveratrol (50 mg–3000 mg per day), patient characteristics (sex, age, or body weight), diet, and lifestyle. This makes it difficult to identify the effects of resveratrol.

In comparison to the results reported in animal studies, the efficacy of this polyphenolic compound in decreasing hepatic steatosis in humans is rather weak. None of the studies reported in this review showed a significant reduction in hepatic lipid levels after resveratrol supplementation [112,113].

During a clinical trial, obese patients with MASLD were given either 3000 mg of resveratrol per day or a placebo for 8 weeks. The trial results showed that this phenolic compound did not enhance liver conditions, plasma lipid profile, insulin resistance or abdominal fat [114]. In a clinical trial involving patients with MASLD, either resveratrol 500 mg or a placebo was administered daily for 12 weeks. The patients followed a complete nutrition plan and increased physical activity at the same time. Both groups showed improvement in hepatic steatosis and function, inflammatory markers, and anthropometric parameters. However, the resveratrol group showed a more significant reduction in inflammatory mediators and hepatic steatosis [90]. In a separate clinical trial, the administration of resveratrol twice daily for three months improved insulin resistance, glucose, and lipid metabolism when compared to placebo patients. Moreover, the resveratrol group had decreased plasma concentrations of cytokeratin 18, TNF-α, and fibroblast growth factor 21, while adiponectin concentrations were higher [115].

Later human studies included high doses of resveratrol (1.5 g daily) given for 6 months to overweight people with histological MASLD. This treatment induced marginal improvements in liver damage, but no differences were observed compared with the control group [116]. Resveratrol supplementation at a dose of 600 mg for 12 weeks in adult subjects with MASLD also failed to show significant beneficial effects. There were no observed improvements in serum liver enzymes or markers of oxidative stress, such as oxidized LDL or glutathione peroxidase and erythrocyte superoxide dismutase [117].

The same authors compared the effects of 600 mg of resveratrol per day for 12 weeks with a low-calorie diet in people with MASLD. The calorie-restricted diet improved anthropometric parameters, lipid profile, and serum transaminases. In contrast, the resveratrol group only showed improvements in weight loss. In addition, there was no effect on hepatic steatosis or circulating SIRT-1 in either group [118]. In a randomized clinical trial, there was also no significant effect of resveratrol supplementation (150 mg daily for 12 weeks) on liver fat levels and cardiometabolic risk parameters compared with the placebo group [119].

The latest research [120] included 82 patients randomly divided into two groups. Patients in the first group were given a mixture of 150 mg of resveratrol and 250 mg of δ-tocotrienol. The patients in the second group received the placebo (cellulose) twice daily for 24 weeks. The findings found that daily supplementation with a mixture of resveratrol and δ-tocotrienol can improve the liver condition by the upregulation of miRNA-130b-5p, which is involved in inflammation and central obesity. Moreover, this mixture may enhance miRNA-221-5p, which is involved in insulin resistance. Additionally, this supplementation resulted in the downregulation of miRNA 122, which improved dyslipidemia.

Another clinical trial evaluated the effects of resveratrol (at a dose of 500 mg three times a day for 6 months) and placebo in patients with MASLD. It was found that liver fat content did not change significantly compared to the placebo group. It also showed no significant effect on the kinetics of basal and insulin-dependent low-density lipoprotein, glucose, and palmitate [121]. Another human study investigated the effects of micronized resveratrol at doses of 50 or 200 mg per day (6 months) in patients with MASLD. This resveratrol supplement, in contrast, reduced insulin resistance, hepatic lipid accumulation and serum liver enzymes [122]. Due to the ambiguity of the results of such studies, it is difficult to confirm the positive effect of resveratrol in treating patients with non-alcoholic fatty liver disease. Thus, it is imperative to conduct additional clinical trials with a substantially greater number of subjects. The results of the resveratrol administration to MASLD patients in clinical trials are presented in Table 3.

The results of studies on pharmacological options for the treatment of MASLD are inconclusive. Currently, the best way to manage MASLD is through lifestyle interventions to achieve weight loss [123]. A reduction of 7% to 10% in body weight, achieved through energy restriction and regular physical activity, is associated with histological improvement resolution of liver fat, fibrosis, and inflammation [124].

The Mediterranean diet is a dietary pattern that has a beneficial effect on the treatment of metabolic syndromes, cancer, and cardiovascular diseases [125]. It may vary among countries and regions, but the common Mediterranean diet consists of primarily consuming olive oil, vegetables, fresh fruit, unrefined cereals, and nuts. Additionally, white meat, fish, legumes, and red wine (resveratrol) can be consumed in moderation, while red meat and sweets should be limited. The diet is characterized by low consumption of saturated fat and cholesterol and high consumption of monounsaturated fatty acids, complex carbohydrates, and fiber.

Research has confirmed that the Mediterranean diet is effective in reducing the risk of CVD, cancer, obesity, and type 2 diabetes [123,126]. Studies suggest that the diet’s ability to reduce the risk of MASLD is due to the nutraceutical effect of bioactive compounds and phytochemicals with antioxidant and anti-inflammatory properties, such as fibers, monounsaturated and omega-3 fatty acids, and phytosterols. MASLD is associated with visceral obesity, insulin resistance, dyslipidemia, and chronic inflammation. Therefore, the Mediterranean diet may improve MASLD due to its antioxidant and anti-inflammatory effects, lipid-lowering effects, and gut-microbiota-mediated production of metabolites.

## 4. Conclusions and Perspectives

Non-alcoholic fatty liver disease is strongly linked with progressive hepatic disease and can cause many cardiometabolic diseases. Due to the lack of specific therapies against MASLD, it is necessary to look for new alternatives, such as the use of bioactive phytochemicals. After analyzing the reported results, it can be stated that in vitro and animal studies have shown promising antisteatotic, antioxidant and anti-inflammatory effects of resveratrol in liver cells. These actions are related to inhibition of the NF-κB pathway and activation of the SIRT-1 and AMPK pathways. Furthermore, numerous studies have indicated gut dysbiosis in patients with non-alcoholic fatty liver disease. Preliminary studies have suggested that resveratrol may have a positive effect on normalizing the gut microbiome, maintaining gut barrier integrity, and ameliorating gut inflammation.

However, when these experimental protocols are transferred to humans, the findings achieved in the clinical studies are ambiguous. Thus, resveratrol does not significantly alleviate liver steatosis in patients, nor does it show a significant reduction in any of the parameters that occurred in non-alcoholic fatty liver disease. These findings that are incompatible with animal studies may be caused by low homogeneity between trials (dose and time of treatment), as well as the low bioavailability of this polyphenol and the variability of the population in relation to animal models. Further and more extensive trials are required to establish the effectiveness of resveratrol and to gain a better understanding of the correlation between lifestyle and diet and the efficacy of this polyphenol. It is possible that new forms of administration, such as nanoparticles, could increase resveratrol bioavailability. Additionally, it is necessary to test whether resveratrol not only contributes to the reversal of MASLD but also prevents its onset and progression into more aggressive forms of pathology. Therefore, longer-term studies with larger numbers of patients are needed to determine the efficacy of resveratrol in the context of MASLD.

## Figures and Tables

**Figure 1 ijms-25-03746-f001:**
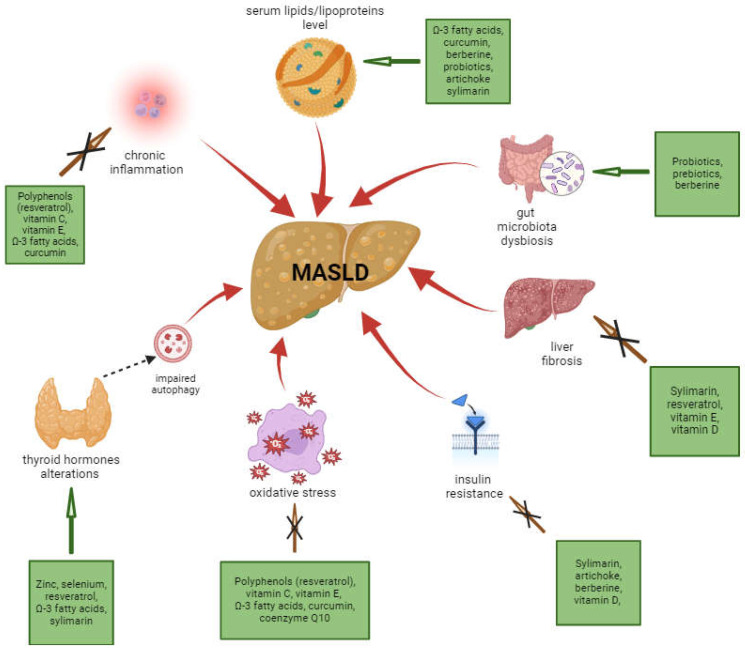
Main targets of nutraceutical improvement of MASLD. Modified from figures in Rizzo et al. [18]. For further explanations of potential mechanisms of MASLD development, please see the text below.

**Figure 2 ijms-25-03746-f002:**
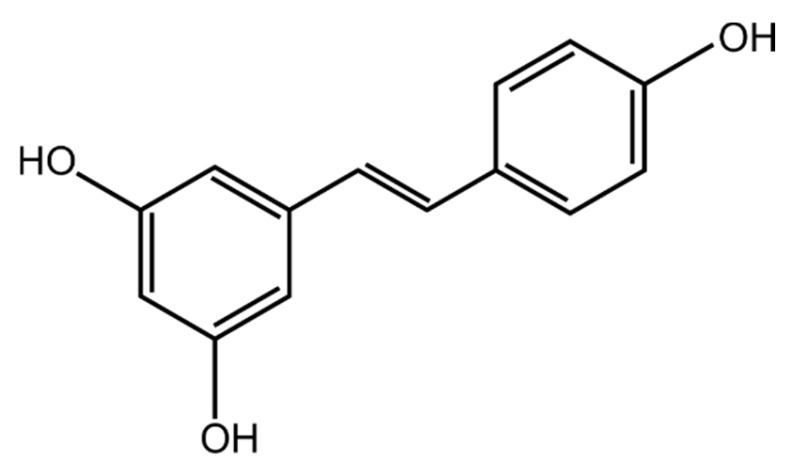
Chemical structure of resveratrol.

**Figure 3 ijms-25-03746-f003:**
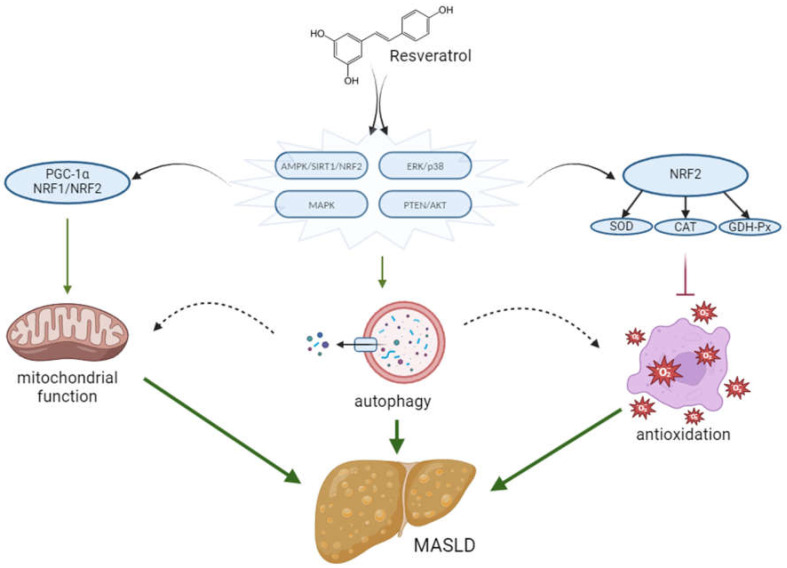
Molecular mechanisms of resveratrol action in the treatment of MASLD. For further explanations, please see the text above.

**Table 1 ijms-25-03746-t001:** Results of the resveratrol administration on hepatic steatosis in in vitro models.

Model	Resveratrol Dose	Mechanism/Results	References
HepG2	80 and 100 μM, 4 weeks	↑ cell viability, autophagy	[89]
HepG2	1–10 μM	↓ ROS production, TNF-α ↑ mitochondrial biogenesis, ↑ mitochondrial respiratory subunits expression, ↑ MnSOD, UCP-2	[86,90]
HepG2	10 and 20 μM. 24 h	↓ steatosis and oxidative stress	[84,85]
HepG2	Resveratrol-loaded D, L-lactide-coglycolide acid	↓ TG accumulation ↑ lipolysis	[91]

**Table 2 ijms-25-03746-t002:** Results of the resveratrol administration on hepatic steatosis in animal models.

Model	Resveratrol Dose	Mechanism/Results	References
Rats	4 mg/kg/day, 7 days	↓ TNF ↓ IL-6 mRNA ↓ CD68+ Kupffer cells recruited in the liver ↓ TIMP-1 ↓ collagen Iα1 mRNA expression ↑ Ki67-positive hepatocytes ↓ cholestatic damage ↓ fibrotic tissue deposition	[82]
Rats	50 and 100 mg/kg/day, 3 weeks	↓ lipid accumulation amelioration of LDL receptor	[94]
Rats	30 mg/kg/day, 60 days	↓ body weight, transaminases ↓ insulin, adipogenic-related genes ↓ TNF-α, IL-6 and NF-κB	[95]
Rats	100 mg/kg/day, 8 weeks	↓ liver steatosis ↓ triglycerides ↑ hepatic mitochondrial biogenesis ↑ UCP-2 expression	[96]
Rats	10 mg/kg/day, 4 weeks	↑ Increase in SOD ↑ GPH-Px ↑ CAT ↓ NOS activity ↓ fatty acid deposition in liver	[97]
Rats	200 mg/kg/day	↑ activation of the AMPK/SIRT1 axis ↓ fatty acid deposition in liver	[100]
Rats	50 mg/kg/day, 120 days	lipid metabolism normalize ↓ oxidative stress ↑ SIRT-1 and leptin expression	[101]
Rats	100 mg/kg/day, 12 weeks	↓ steatosis ↓ insulin resistance ↓ oxidative stress ↓ IL-6 and TNF-α ↑ SIRT1 pathway	[102]
Rats	10 mg/kg/day, 28 days	↑ liver histology, lipid profile and glycemia ↓ leptin	[106]
Rats	20 mg/kg/day, 8 weeks	Liver steatosis amelioration ↑ insulin sensitivity CPT-1 and UCP-2 normalization ↓ SREBPs-1 and -2 cleavage ↓ ROS ↓ pancreatic lipase	[107]
Rats	500 mg 6 months	AMPK/SIRT1/Nrf2, ERK/p38, MAPK and PTEN/Akt signaling pathways inducing autophagy through an mTOR-dependent or TFEB-dependent pathway Synthesis of antioxidant molecules and the expression of related genes involved in the biogenesis of mitochondrial energy	[108]
Rats	10 to 20 mg/kg/day, 8 weeks	↓ MDA ↑ GPH-Px and SOD ↓ mRNA expression of inflammatory mediators (NO, TNF-α, IL-1)	[109]
Rats	20 mg/kg/day, 3 weeks	↓ MDA ↑ GPH-Px and SOD ↓ mRNA expression of inflammatory mediators (NO, TNF-α, IL-1) ↓ liver tissue infiltration of inflammatory cells and fibrosis deposition	[110]
Rats	20 mg/kg/day, 3 days per week	↓ MDA ↑ GPH-Px and SOD ↓ mRNA expression of inflammatory mediators (NO, TNF-α, IL-1) ↓ liver tissue infiltration of inflammatory cells and fibrosis deposition	[111]
Mice	30 mg/kg/day, 60 days	↓ lipid accumulation ↓ TNF-α, IL- 6 and IL-1β, and NF-κB ↑ AMPKα phosphorylation and SIRT-1	[80]
Mice	2 and 20 mg/kg/day, 4 weeks	liver steatosis unchanged	[99]
Mice	resveratrol-curcumin hybrid 10 and 20 mg/kg every 2 days, 12 weeks	↓ lipid accumulation, hepatic inflammation, fibrosis	[101]

**Table 3 ijms-25-03746-t003:** Results of the resveratrol administration to MASLD patients in clinical trials.

Patients	Resveratrol Dose	Mechanism/Results	References
20 MASLD patients (men)	3000 mg/day, 8 weeks	No changes in liver, lipids, or insulin	[114]
50 MASLD patients (men and women)	500 mg/day, 12 weeks, diet, exercise	↓ ALT ↓ NF-κB ↓ cytokeratin-18 ↓ hepatic steatosis grade	[90]
60 MASLD patients (men and women)	150 mg/twice day, 3 months	↑ glucose and lipid metabolism ↑ insulin resistance ↓ NF-α and cytokeratin 18 ↓ fibroblast growth factor 21 ↑ adiponectin	[115]
90 MASLD patients (men and women)	600 mg/day, 12 weeks, calorie-restricted diet	weight loss and body mass index improvements	[118]
112 MASLD patients (men and women)	150 mg/day, 12 weeks	liver fat content, cardiometabolic risk parameters, insulin resistance unchanged	[119]
16 MASLD patients (men)	500 mg/3 times daily 6 months	liver fat content unchanged	[121]
44 MASLD patients (men and women)	50 and 200 mg/day, 6 months	↓ hepatic lipid accumulation, enzymes ↓ insulin resistance	[122]

## Data Availability

The data presented in this study are available on request from the corresponding author.

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
