# Peer review of "Does Resveratrol Improve Metabolic Dysfunction-Associated Steatotic Liver Disease (MASLD)?"

_ijms, 2024, doi:10.3390/ijms25073746_

Round 1

Reviewer 1 Report

Comments and Suggestions for Authors

This is an interesting review, however, I have some suggestions for the Authors 

-NAFLD is an old definition of the disease. Since June 2023, the hepatology community has introduced the term MAFLD. I, therefore, suggest updating the text accordingly, citing the document https://doi.org/10.1016/j.jhep.2023.06.003.

-the first part on the pathogenesis of NAFLD is very long and not entirely functional to the specific topic of the review, which would be the role of resveratrol in the improvement of NAFLD. Please consider reducing it.

-Did the authors carry out a systematic search of clinical trials with resveratrol?

​

Comments on the Quality of English Language

Please check the text for some typos

Author Response

The authors would like to thank the Reviewer for the valuable comments which have helped to improve the quality of the paper. All the changes and corrections are highlighted in the text.

This is an interesting review, however, I have some suggestions for the Authors. 

-NAFLD is an old definition of the disease. Since June 2023, the hepatology community has introduced the term MAFLD. I, therefore, suggest updating the text accordingly, citing the document https://doi.org/10.1016/j.jhep.2023.06.003.

Thank you for your rightful comment. We have amended the definition of disease in line with the latest nomenclature.

-the first part on the pathogenesis of NAFLD is very long and not entirely functional to the specific topic of the review, which would be the role of resveratrol in the improvement of NAFLD. Please consider reducing it. Przemek

Thank you for comment The first part of the article has been significantly shortened.

-Did the authors carry out a systematic search of clinical trials with resveratrol?

A non-systematic review, using Preferred Reporting Items for Systematic Reviews and Meta - Analyses (PRISMA) guidelines (Page, M.J.; McKenzie, J.E.; Bossuyt, P.M.; Boutron, I.; Hoffmann, T.C.; Mulrow, C.D.; Shamseer, L.; Tetzlaff, J.M.; Akl, E.A.; Brennan, S.E.; et al. The PRISMA 2020 Statement: An Updated Guideline for Reporting Systematic Reviews. BMJ 2021, 372, n71, doi:10.1136/bmj.n71.) was conducted. As the literature search engine, the browsers in the Scopus, PubMed, Web of Science databases and ClinicalTrials.gov register were used. The following inquiries were used: “NAFLD and polyphenols”, “NAFLD and resveratrol”, “NAFLD and ‘in vitro’ studies”, “NAFLD and ‘clinical trials’”, “NAFLD and ‘animal models’”, “resveratrol and ‘hepatic steatosis’”, “resveratrol and ’non-alcoholic fatty liver disease’”. Documents published from 1980 to October 2023 were included. In order to qualify work for the review the following exclusion criteria were applied: works do not contain original data e.g. reviews, comments, works which have not been independently peer-reviewed e.g. confer-ence papers, letters to editor, pre-prints etc., works in other than English language, papers published before 1980 year

The search was conducted as follows: Prof. Anna Oniszczuk (A.O.) and Dr. Kamila Kasprzak-Drozd (K.K-D.) identified relevant studies by reading the abstracts and searching for additional studies through the reference lists of the selected papers. Then, A.O. and K.K-D. independently reviewed the studies by checking the titles and abstracts of the articles and deciding whether to include each article in accordance with exclusion criteria described above. Disaccords between reviewers have been resolved by discussion and consensus or by arbitration (T.O.).

Reviewer 2 Report

Comments and Suggestions for Authors

General comments

The main part of the article should be focused on resveratrol and its role in prevention and treatment of NAFLD. In accordance with the mechanisms of NAFLD given in section 2.1 – 2.5 , resveratrol effects should also be classified in such a way and related to the specific mechanism of its action. I strongly recommend to authors to reorganize the paper and include separate part (reorganize section 3.3, 3.4, 3.5) in order to discuss the research ideas of this paper and summarize the results for each mechanism of action. Table could be useful in order to summarize the results.

Specific comments

-Abstract needs to be rewritten in order to better reflect the results of in vitro and in vivo studies.

-Considering the new terminology, authors should think about changing the title.

- The introduction section, as the beginning of the article, only needs to briefly introduce the research background and research ideas of the topic of this article. Hence, the introduction part should be reorganized. There is a lot of repetition and MASLD is a new term for NAFLD -metabolic dysfunction associated steatotic liver disease. Authors should give brief explanation about the changes in terminology and reasons for that. For example, the sentence given in line 42 is more appropriate for the beginning of the introduction section. In line 54-please indicate diseases and avoid using the term „civilization disease”. Sentence given in line 62-63 should be rewritten.

-Mediterranean diet should be addressed in few lines as well. It is a gold standard in preventive medicine especially NAFLD because provides food micronutrients which include polyphenols, especially resveratrol from grape and red wine as well as olive oil (https://pubmed.ncbi.nlm.nih.gov/31215262/  https://pubmed.ncbi.nlm.nih.gov/28805669/  https://doi.org/10.1177/1934578X1801300923)

-Considering the fact that the title of this review paper is “Does resveratrol improve non-alcoholic fatty liver disease?” presented Figure 1 should be modified and metabolism of lipids as one of the main targets should also be include.

-Section 2.3 should contain strains that have been correlated with NAFLD. For example, enrichment of Eubacterium, Bacteroides, Escherichia, have been documented in children with NAFLD.

Comments on the Quality of English Language

Minor improvements, especially in abstract and introduction are needed. 

Author Response

The authors would like to thank the Reviewer for the valuable comments which have helped to improve the quality of the paper. All the changes and corrections are highlighted in the text.

General comments

The main part of the article should be focused on resveratrol and its role in prevention and treatment of NAFLD. In accordance with the mechanisms of NAFLD given in section 2.1 – 2.5, resveratrol effects should also be classified in such a way and related to the specific mechanism of its action. I strongly recommend to authors to reorganize the paper and include separate part (reorganize section 3.3, 3.4, 3.5) in order to discuss the research ideas of this paper and summarize the results for each mechanism of action. Table could be useful in order to summarize the results.

Specific comments

-Abstract needs to be rewritten in order to better reflect the results of in vitro and in vivo studies.

Thank you for comment, the abstract has been completely revised.

-Considering the new terminology, authors should think about changing the title.

Thank you for your rightful comment. We have amended the definition of disease in line with the latest nomenclature. We have also updated the title of the review.

- The introduction section, as the beginning of the article, only needs to briefly introduce the research background and research ideas of the topic of this article. Hence, the introduction part should be reorganized. There is a lot of repetition and MASLD is a new term for NAFLD -metabolic dysfunction associated steatotic liver disease. Authors should give brief explanation about the changes in terminology and reasons for that. For example, the sentence given in line 42 is more appropriate for the beginning of the introduction section. In line 54-please indicate diseases and avoid using the term „civilization disease”. Sentence given in line 62-63 should be rewritten.

Introduction has been rewritten, as suggested by the Reviewer,

-Mediterranean diet should be addressed in few lines as well. It is a gold standard in preventive medicine especially NAFLD because provides food micronutrients which include polyphenols, especially resveratrol from grape and red wine as well as olive oil (https://pubmed.ncbi.nlm.nih.gov/31215262/  https://pubmed.ncbi.nlm.nih.gov/28805669/  https://doi.org/10.1177/1934578X1801300923)

Thank you for your valuable comment. The section 3.5. has been augmented with information on Mediterranean diet.

-Considering the fact that the title of this review paper is “Does resveratrol improve non-alcoholic fatty liver disease?” presented Figure 1 should be modified and metabolism of lipids as one of the main targets should also be include.

Thank you for your valuable comment. A necessary modifications including lipid metabolism combined with potential nutraceuticals have been implemented in Figure 1.

-Section 2.3 should contain strains that have been correlated with NAFLD. For example, enrichment of Eubacterium, Bacteroides, Escherichia, have been documented in children with NAFLD.

Thank you for your valuable comment. The section 2.3 has been augmented with studies on microbiome alterations in NAFLD (MASLD). The most important phyla/genera according to literature data have been also noticed in text.

Reviewer 3 Report

Comments and Suggestions for Authors

This paper is a review of the effects of resveratrol in MASLD for humans and animal models.  Although, animal models do show some support for resveratrol in alleviating MASLD, the recapitulation of these results in humans has been disappointing.  This manuscript does a good job with covering the subject matter, but there are improvements to be made.

1.       The first seven pages review NAFLD (MASLD) so I am not sure if this amount of review material is even necessary.  It may be better to summarize most of this to provide a framework of understanding that is pertinent to resveratrol. 

2.      NAFLD should be changed to MASLD according to the new nomenclature that was established last year.

3.      Line 37/38 should be reworded as it seems that steatosis is established when about 5% of hepatocytes are symptomatic of this disorder.  The assertion is that 5% of the weight of the liver is lipid material which is the definition of simple steatosis.

4.      Line 91, “a many of NPs” is not good grammar.

5.      Line 135:  Re-word and avoid using “overrun”

6.      Lines 140-151:  Please use the word “lipotoxicity” to describe this process.

7.      Line 180:  Explain why fructose is detrimental. 

8.      Line 213:  Is butyrate the only SCA or is this an example of that chemical group?

9.      2.3.3 Amino acids:  This section should either be developed better or deleted altogether.

10.  Section 3.3:  The HepG2 model is not very good for modeling resveratrol metabolism since this cell line is from a carcinoma and does not resemble hepatocytes.  This should be emphasized in the review as a commentary.

11.  Line 436 is missing a word or two.

12.  Line 539:  They injected resveratrol?  What route (this makes a difference).  Is this a one-off instance or should Table 1 be expanded to include a column of “dosage and route”?

13.  Perhaps another table should be included for human clinical trials and also include the trial number.

14.  Line 626 change “researches” to “experimental protocols”

Comments on the Quality of English Language

Some minor grammatical mistakes that are easily fixable.

Author Response

The authors would like to thank the Reviewer for the valuable comments which have helped to improve the quality of the paper. All the changes and corrections are highlighted in the text.

This paper is a review of the effects of resveratrol in MASLD for humans and animal models.  Although, animal models do show some support for resveratrol in alleviating MASLD, the recapitulation of these results in humans has been disappointing.  This manuscript does a good job with covering the subject matter, but there are improvements to be made.

  1. The first seven pages review NAFLD (MASLD) so I am not sure if this amount of review material is even necessary.  It may be better to summarize most of this to provide a framework of understanding that is pertinent to resveratrol. 
    Thank you for comment The first part of the article has been significantly shortened.
  2. NAFLD should be changed to MASLD according to the new nomenclature that was established last year.

Thank you for your rightful comment. We have amended the definition of disease in line with the latest nomenclature. We have also updated the title of the review.

  1. Line 37/38 should be reworded as it seems that steatosis is established when about 5% of hepatocytes are symptomatic of this disorder. The assertion is that 5% of the weight of the liver is lipid material which is the definition of simple steatosis.

The poorly worded phrase has been corrected.

  1. Line 91, “a many of NPs” is not good grammar.

Thank you for comment, the phrase has been corrected

  1. Line 135:  Re-word and avoid using “overrun”

Thank you for comment, the phrase has been corrected

  1. Lines 140-151:  Please use the word “lipotoxicity” to describe this process.

Thank you for comment, the word “lipotoxicity” was used to define the process described

  1. Line 180:  Explain why fructose is detrimental. 

Thank you for comment, an explanation in the text has been implemented. High fructose diet has been connected in studies on mice with alterations in certain microbiome genus, which resulted in increased gut permeability and elevated pro-inflammatory molecules expression in serum.

  1. Line 213:  Is butyrate the only SCA or is this an example of that chemical group?

Thank you for comment, the phrase has been corrected and disambiguated. To our best knowledge, butyrate is the most extensively studied SCA in MASLD and is given as a sample of this group of compounds.

  1. 2.3.3 Amino acids:  This section should either be developed better or deleted altogether.

Thank you very much for your kind comment, this section has been delated.

  1. Section 3.3:  The HepG2 model is not very good for modeling resveratrol metabolism since this cell line is from a carcinoma and does not resemble hepatocytes.  This should be emphasized in the review as a commentary.

Thank you very much for your kind comment. The characterisation of HepG2 cells is included in the section entitled "Effect of resveratrol on liver condition - in vitro studies"

  1. Line 436 is missing a word or two.

Thank you for your comment. The text has been corrected.

  1. Line 539:  They injected resveratrol?  What route (this makes a difference).  Is this a one-off instance or should Table 1 be expanded to include a column of “dosage and route”?

The researchers used an intraperitoneal injection and this was the only single case of this route of administering resveratrol. The information has been added to the text.

  1. Perhaps another table should be included for human clinical trials and also include the trial number.

Thank you for comment. We have included the table with human clinical trials.

  1. Line 626 change “researches” to “experimental protocols”

Thank you for your comment. The phrase has been corrected.

Reviewer 4 Report

Comments and Suggestions for Authors

The authors of the review are thanked for the quality of the work and the choice of the subject relating to the available data on the therapeutic effects of resveratrol against NAFLD. It is known that  nonalcoholic fatty liver disease (NAFLD) is the most common liver disorders worldwide. Kasprzak-Drozd and collaborators describes the in vitro and animal model studies showing promising results with resveratrol. On the other hand, the authors of the presenting review focus on clinical trials, where  resveratrol does not significantly improve any of the NAFLD parameters. In conclusion, the authors suggest that more studies involving a larger number of patients and over a longer period of time are needed. The manuscript entitled “Does resveratrol improve non-alcoholic fatty liver disease?” is a good study, scientifically valid, well executed, and deserve some space in the journal. After reading the manuscript thoroughly, I have no comments to the authors. I believe the manuscript is very good and can be published in present form.

Author Response

We would like to thank the Reviewer for kind comments and appreciation of our review article.

Round 2

Reviewer 1 Report

Comments and Suggestions for Authors

The Authors have improved the paper.

Please check Figure 3, NAFLD should be modified in the image.  

Author Response

The Authors have improved the paper. Please check Figure 3, NAFLD should be modified in the image.  

Thank you for comment. The figure has been modified.

Reviewer 2 Report

Comments and Suggestions for Authors

General comments

The authors made significant improvements. However, some minor corrections have to be done.

Specific comments

1. Lines 80-88: The sentence should be rewritten.

2. Line 93: Full stop is missing.

3. Table 3, gender of patients should be indicated (women and/or man) in column related to patients

Author Response

General comments

The authors made significant improvements. However, some minor corrections have to be done.

Specific comments

  1. Lines 80-88: The sentence should be rewritten.

Thank you for comment. Sentences have been rewritten.

  1. Line 93: Full stop is missing.

Thank you for comments, this has been corrected.

  1. Table 3, gender of patients should be indicated (women and/or man) in column related to patients

Thank you for comments. Gender of patients was added to the table 3.

Reviewer 3 Report

Comments and Suggestions for Authors

Extensive revisions have been made and the paper looks to be more focused and reads better than previously.  One misspelled word on line 813, but overall, much better and ready for publication.

Comments on the Quality of English Language

Basic proofing for misspelled words and grammar required, but no major concerns.

Author Response

Extensive revisions have been made and the paper looks to be more focused and reads better than previously.  One misspelled word on line 813, but overall, much better and ready for publication.

Thank you for comments, this has been corrected.